# Continuous plate subduction marked by the rise of alkali magmatism 2.1 billion years ago

He Liu [1,2,3], Wei-dong Sun[1,2,3], Robert Zartman[4] & Ming Tang[5]

Over the Earth's evolutionary history, the style of plate subduction has evolved through time due to the secular cooling of the mantle. While continuous subduction is a typical feature of modern plate tectonics, a stagnant-lid tectonic regime with localized episodic subduction likely characterized the early Earth. The timing of the transition between these two subduction styles bears important insights into Earth's cooling history. Here we apply a statistical analysis to a large geochemical dataset of mafic rocks spanning the last 3.5 Ga, which shows an increasing magnitude of alkali basaltic magmatism beginning at ca. 2.1 Ga. We propose that the rapid rise of continental alkali basalts correlates with an abruptly decreasing degree of mantle melting resulting from the enhanced cooling of the mantle at ca. 2.1 Ga. This might be a consequence of the initiation of continuous subduction, which recycled increasing volumes of cold oceanic crust into the mantle.

[1] Center of Deep-Sea Research, Institute of Oceanology, Chinese Academy of Sciences, 7 Nanhai Road, 266071 Qingdao, China. [2] Laboratory for Marine Mineral Resources, Qingdao National Laboratory for Marine Science and Technology, 266237 Qingdao, China. [3] Center for Ocean Mega-Science, Chinese Academy of Sciences, 7 Nanhai Road, 266071 Qingdao, China. [4] Department of Earth Atmospheric and Planetary Sciences, Massachusetts Institute of Technology, Cambridge, MA 02139, USA. [5] Department of Earth, Environmental and Planetary Sciences, Rice University, Houston, TX 77005, USA. Correspondence and requests for materials should be addressed to W.-d.S. (email: weidongsun@qdio.ac.cn)

Earth's tectonics evolved through multiple phases coupled with its cooling history. In the early stage, the Earth's geodynamic model was dominated by a stagnant-lid vertical tectonic regime[1–6]. The recycling of oceanic plates was likely performed by episodic subduction[2,4,7–9]. Such episodic subduction was mainly recognized by numerical modeling as the subducting plate underwent frequent slab break-off in the hot mantle (ca. 175–250 °C hotter than the present mantle)[7,9]. As a result of the secular cooling of the mantle, the subduction style was later transitioned into the self-sustaining continuous subduction[2,9,10]. The continuous subduction that dominates at the modern convergent plate boundaries, is characterized by long-lived recycling of oceanic plate into the mantle along the subduction zone[7,9,10].

The style of plate subduction has profound influences on orogenic processes, continent evolution, and mass exchange between the shallow and deep Earth. Although the onset of plate tectonics has been highly contested with estimates ranging from >4 Ga to <1 Ga[3,5,11–25], little is known about the pace of subduction style evolution, particularly the transition from episodic to continuous plate subduction. Based on the abundantly preserved Neoproterozoic and younger ophiolites, the modern-style continuous subduction has been inferred to commence at ca. 1.0 Ga in the Neoproterozoic era[11,12]. However, the Jormua ophiolite in Finland and Purtuniq ophiolite in Canada support the generation and emplacement of such rocks at ca. 2.0 Ga in the Paleoproterozoic[26,27]. Furthermore, more recent studies on the newly discovered ~1.83 Ga eclogite in the Trans-Hudson orogen of North America[28] and the ~1.82 Ga eclogite xenolith in a Paleoproterozoic carbonatite unit in North China[29] imply that local episodes of deep subduction had already occurred before ca. 1.8 Ga. The previously recognized 2.2–2.0 Ga blueschist-facies metamorphic rocks from the West African craton also suggest that modern-style subduction operated, at least locally, as early as the Paleoproterozoic[30]. A more recent compilation of geochemical and palaeomagnetic data suggests that the change from the pre-plate tectonic stagnant-lid setting to sustained plate tectonics transitioned over a long period from 3.2 to 2.5 Ga[4].

An important consequence of continuous subduction is the enhanced cooling of the mantle caused by the increasing volume of subducting oceanic crust[31–33], which, in turn, would result in lower degrees of mantle partial melting[32]. The decreasing degree of mantle melting should lead to a reduction of komatiites[33] and an increase in continental alkali basalts. Here we apply a statistical analysis to the geochemical data of mafic igneous rocks across two typical modern arc-continent tectonic systems to test the geochemical behaviors of several specific incompatible elements from their magmatic arcs towards their continental interiors. Then we evaluate ~55,000 mafic igneous rocks spanning the last 3.5 Ga to track the initiation and consequence of continuous plate subduction. The start time of continuous plate subduction is estimated as ca. 2.1 Ga.

## Results

**High-field strength elements systematics.** The High-field strength elements (HFSE; e.g. Nb, Ta, Ti), as well as phosphorus (P), are relatively fluid-immobile elements, which are predominantly retained by insoluble minerals (e.g. rutile, apatite, etc.) during the subduction dehydration process[34–36]. This retention process accounts for the relatively low HFSE and P concentrations of arc mafic rocks. By contrast, mafic magmas that erupt in the interior of continents are dominated by sodic alkali basalts with lesser amounts of potassic alkali basalts and tholeiitic basalts[37]. These intracontinental alkali basalts are predominantly characterized by high HFSE and P concentrations. They have primitive-mantle normalized trace element patterns similar to those of oceanic island basalts (OIB)[37].

The magmatic sources of continental alkali basalts are considered to be derived from melting in upwelling sublithospheric mantle or from lithospheric mantle that has been metasomatized by asthenosphere-derived melts[37]. Whether the mantle melting is induced by actively upwelling high-temperature plumes or by passively upwelling asthenosphere as a consequence of lithospheric extension in both localized continental rifts (e.g. East African Rift, Baikal, etc.) and broader continental regions (e.g. eastern China, southern Argentina, western US, Southeast Asia, south-eastern Australia, etc.)[37–42], the generation of alkali basaltic magma usually requires low degrees of mantle melting[37,43,44].

We chose two typical sites of modern arc-continent tectonic systems, the Southern Andes of South America and East Asia, as examples to illustrate the difference in HFSE and P systematics between arc mafic rocks and continental alkali basalts with the current thermal state of the mantle. Geochemical data for arc and intracontinental mafic rocks were extracted from the EarthChem rock database[45]. In the Southern Andes as well as the northern tip of the Antarctic Peninsula, arc mafic rocks near the Pacific coastline contain relatively low HFSE and P concentrations, while the intracontinental mafic rocks, dominated by alkali basalts, contain high HFSE and P concentrations (Fig. 1a–c, e, f). It is also noteworthy that the $SiO_2$ concentration in the majority of arc mafic rocks is >49 wt%, while the intracontinental alkali basalts contain variable $SiO_2$ concentrations from 45 to 52 wt% (Fig. 1d, h). We calculated the average HFSE and P concentrations of the combined arc and intracontinental mafic rocks for each 1 wt% $SiO_2$ interval from 45 to 52 wt% (Fig. 1e–g). The combined HFSE-$SiO_2$ and P-$SiO_2$ compositional trends show that the average HFSE and P concentrations in low-silica mafic rocks (45–49 wt% $SiO_2$) are statistically higher than those in high-silica mafic rocks (49–52 wt% $SiO_2$; Fig. 1e–g; Supplementary Table 1). Similarly, in East Asia (Fig. 2), the arc mafic rocks located on the Kamchatka Peninsula, Kuril Islands, Japanese Islands, Ryukyu Islands, and Izu-Bonin-Mariana Arc have relatively low-HFSE, low-P (Fig. 2a–c, e–g), and high-$SiO_2$ concentrations (mostly 49–52 wt% $SiO_2$; Fig. 2d, h), while the intracontinental alkali basalts in eastern China are significantly more HFSE and P-enriched (Fig. 2a–c, e–g) with more variable $SiO_2$ concentrations (Fig. 2d, h). Consequently, from the combined compositional trends (Figs. 1e–g and 2e–g; Supplementary Tables 1 and 2), we may conclude that with the current thermal state of the mantle the low-silica mafic rocks contain statistically higher average HFSE and P concentrations than the high-silica mafic rocks in the arc-continent tectonic systems.

The above two examples show the typical HFSE-$SiO_2$ and P-$SiO_2$ fractionation existing between arc mafic rocks and continental alkali basalts with the present, or the Phanerozoic (541 Ma to the present), thermal state of the mantle. Although the alkali basaltic rocks were commonly considered as the products of low degrees of mantle melting[37,43], they were rarely seen in the Archean as the mantle temperature then was much hotter than its temperature during the Phanerozoic[13,46], thus favoring higher degrees of mantle melting. In view of this fact, we extracted the relevant geochemical data from the EarthChem database for Phanerozoic and Archean mafic rocks (excluding komatiites and komatiitic basalts) and examined their combined Nb-$SiO_2$, $TiO_2$-$SiO_2$, and P-$SiO_2$ compositional trends. The results show that all the average Nb, Ti, and P concentrations of Phanerozoic mafic rocks decrease as $SiO_2$ concentrations increase from 45 to 52 wt% (Fig. 3; Supplementary Table 3). Again, the average HFSE and P concentrations in low-silica mafic rocks are statistically higher than those in high-silica mafic rocks, which is very similar to the

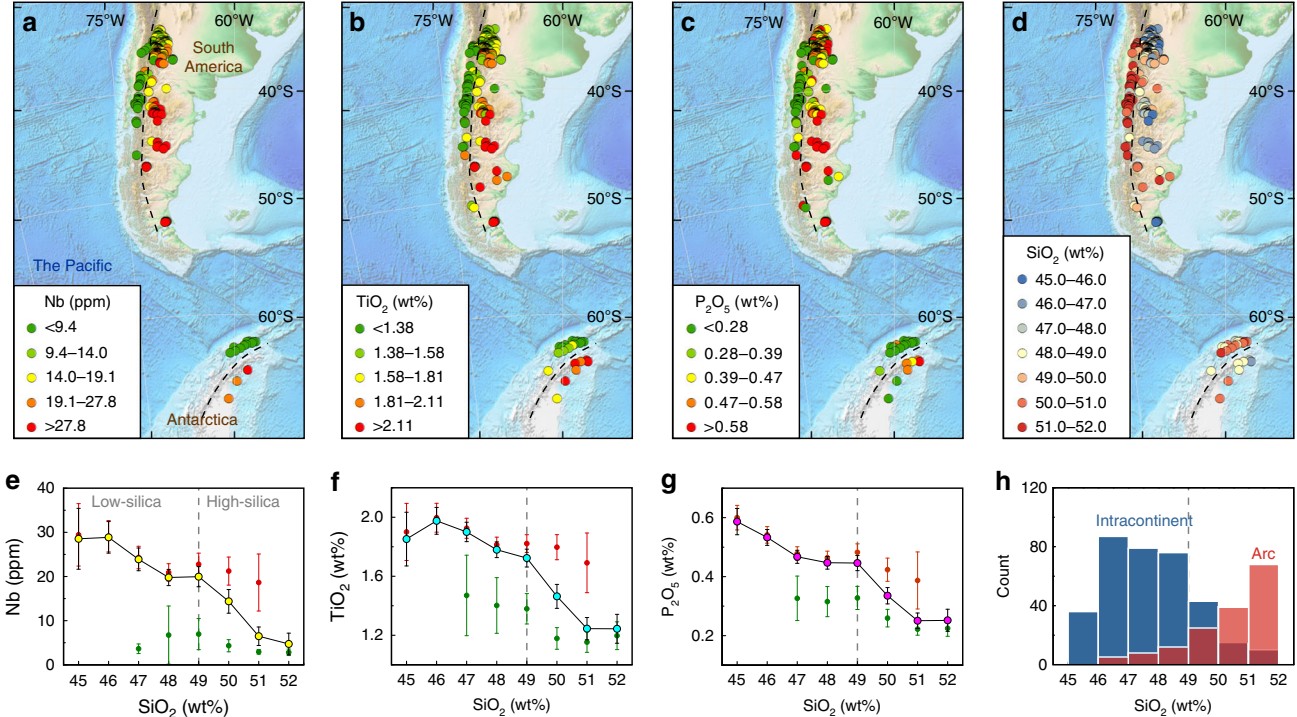

**Fig. 1** Fractionation of Nb, Ti and P-SiO$_2$ of mafic rocks from South America. **a–d** Locations and concentrations of Nb (ppm), TiO$_2$ (wt%), P$_2$O$_5$ (wt%), and SiO$_2$ (wt%), respectively, of mafic rock samples (0–5 Ma ages) from the southern Andes of South America and the Antarctic Peninsula. The black dashed lines divide the rock samples into arc (west) and continental (east) groups. Background relief maps are downloaded from NOAA website (https://www.ngdc.noaa.gov/)[61]. **e–g** Nb-SiO$_2$, TiO$_2$-SiO$_2$, and P$_2$O$_5$-SiO$_2$ compositional trends, respectively, of mafic rocks within the map area. The circles with yellow, cyan, and magenta infills denote the average Nb, TiO$_2$, and P$_2$O$_5$ concentrations, respectively, of all mafic rocks within the map area. Error bars denote two standard errors of the means (s.e.m.). The gray vertical-dashed line shows a proposed subdivision between low-silica (45–49 wt% SiO$_2$) and high-silica (49–52 wt% SiO$_2$) mafic rocks. **h** Histograms showing the frequency of arc (red) and intracontinental (blue) mafic rocks

situations in the Southern Andes and East Asia (Figs. 1 and 2). Most likely it is the extensive Phanerozoic continental alkali basalts from those intracontinental regions as well as continental rifts that have significantly elevated the average HFSE and P concentrations in the low-silica mafic rocks. By contrast, all the Nb-SiO$_2$, Ti-SiO$_2$, and P-SiO$_2$ combined compositional trends of the Archean mafic rocks differ significantly from those of the Phanerozoic mafic rocks (Fig. 3; Supplementary Table 3). Here the average HFSE and P concentrations of low-silica mafic rocks are approximately equal to those in high-silica mafic rocks.

**Determining the rise of alkali basalts.** The different characteristics of Nb-SiO$_2$, Ti-SiO$_2$, and P-SiO$_2$ fractionation between the Archean and the Phanerozoic are attributable to the different mantle temperature[46]. The relatively cold mantle temperatures in the Phanerozoic were more favorable for low degrees of melting and generation of alkali basaltic magmas with high Nb, Ti, and P concentrations and low-silica contents. The hot mantle in the Archean would facilitate higher degree of melting with the production of more tholeiitic basalts with low Nb, Ti, and P concentrations. Therefore, the magnitude of intracontinental alkali basalt eruptions is likely to have been limited in the Archean. We calculated the proportion of alkali basalts in relation to all mafic rocks for every 0.25-Ga time interval. As shown in Fig. 4b, the proportion of alkali basalts is only ~5% in the Archean, whereas in the Phanerozoic, alkali basalts account for >40% of total mafic rock production (exclusive of oceanic rocks). This result corroborates the assumption that the hot mantle in the Archean, or even subsequently, has not contributed to the formation of extensive continental alkali basalts. Such a small proportion of alkali basalts explains why the average Nb, Ti, and P

concentrations in low-silica mafic rocks in the Archean are not elevated (Fig. 1a–c).

We now examine the entire mafic rock database (Supplementary Data 1) to trace the rise of alkali basalts and reconstruct the history of changes in thermal state of the mantle. To quantify the contribution from alkali basalts, we introduce a new geochemical proxy, Diff (HFSE), which reflects the relative enrichments or depletions in HFSE (or P) with increasing SiO$_2$ in mafic rocks:

$$\text{Diff}(\text{HFSE}) = \omega(\text{HFSE})_{\text{low-Si}} - \omega(\text{HFSE})_{\text{high-Si}} \quad (1)$$

The $\omega(\text{HFSE})_{\text{low-Si}}$ is the average HFSE or P concentrations in rocks with 45–49 wt% SiO$_2$, while $\omega(\text{HFSE})_{\text{high-Si}}$ is the average HFSE or P concentrations in rocks with 49–52 wt% SiO$_2$. The secular variations of Diff (Nb), Diff (Ti), and Diff (P) over time are shown in Fig. 4a and Supplementary Tables 4–6. All the Diff (HFSE) vary over a small range around zero from ca. 3.0 to 2.1 Ga, after which they show an abrupt increase and maintain values above zero through the rest of the Earth's history. The rapid increase of Diff (HFSE) from zero to positive values over the 2.1–1.8 Ga time range compellingly reveals that the combined Nb-SiO$_2$, Ti-SiO$_2$, and P-SiO$_2$ compositional trends of mafic rocks were changing from an Archean-like style to a Phanerozoic-like style (Fig. 3). After 1.8 Ga, the Diff (HFSE) vary as positive values with relatively small fluctuations until ca. 0.1 Ga. The enhancement of Diff (Nb) and Diff (P) at the last 0.1 Ga time interval are caused by the over sampled alkali basalts (both high-Ti and low Ti) from the continental rifts (especially from the East African Rift) and volcanic cones in the broad continental areas for the most recent 50 Ma, which are much less preserved before 50 Ma (Supplementary Fig. 1). Thus, it may not have a geological significance.

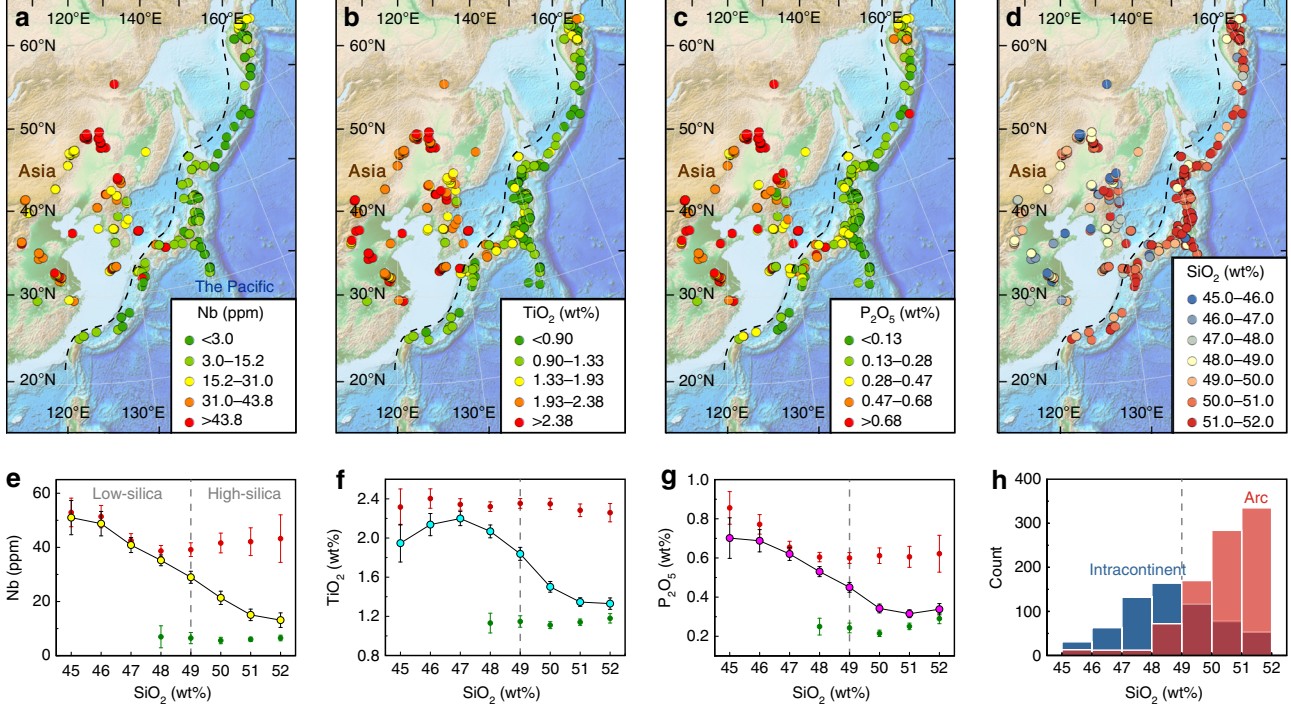

**Fig. 2** Fractionation of Nb, Ti, and P-SiO$_2$ of mafic rocks from East Asia. **a–d** Locations and concentrations of Nb (ppm), TiO$_2$ (wt%), P$_2$O$_5$ (wt%), and SiO$_2$ (wt%), respectively, of mafic rock samples (0–5 Ma ages) from East Asia. The black dashed lines divide the rock samples into arc (east) and continental (west) groups. Background relief maps are downloaded from NOAA website (https://www.ngdc.noaa.gov/)[61]. **e–g** Nb-SiO$_2$, TiO$_2$-SiO$_2$, and P$_2$O$_5$-SiO$_2$ compositional trends, respectively, of mafic rocks within the map area. The circles with yellow, cyan, and magenta infills denote the average Nb, TiO$_2$, and P$_2$O$_5$ concentrations, respectively, of all mafic rocks within the map area. Error bars denote two standard errors of the means (s.e.m.). The gray vertical-dashed line shows a proposed subdivision between low-silica (45–49 wt% SiO$_2$) and high-silica (49–52 wt% SiO$_2$) mafic rocks. **h** Histograms showing the frequency of arc (red) and intracontinental (blue) mafic rocks

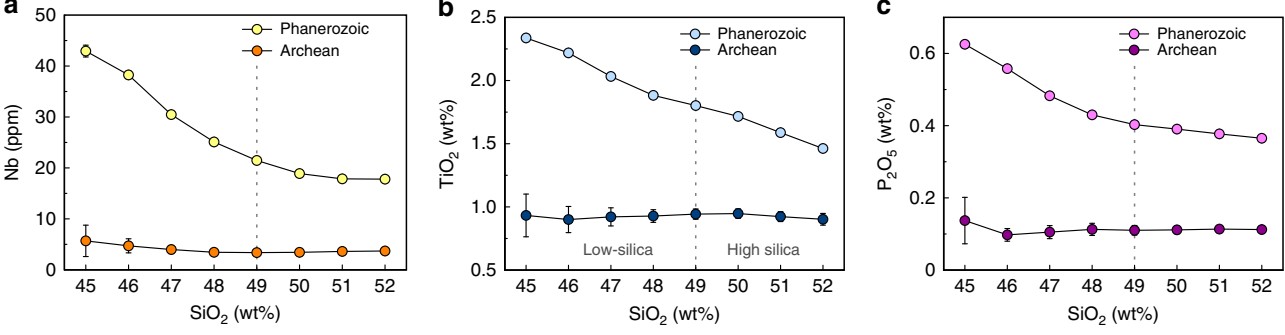

**Fig. 3** Combined Nb, TiO$_2$, and P$_2$O$_5$-SiO$_2$ compositional trends of mafic rocks. **a–c** Diagrams of the combined Nb-SiO$_2$, TiO$_2$-SiO$_2$, and P$_2$O$_5$-SiO$_2$ compositional trends of mafic rocks formed in the Archean and in the Phanerozoic, respectively. The round circles are mean values corresponding to SiO$_2$ concentrations ranging from 45 to 52 wt% (1 wt% SiO$_2$ interval) as estimated by the Bootstrap methods. Error bars denote two standard errors of the mean (s.e.m.). The gray vertical-dashed lines divide the rocks into low-silica (45–49 wt% SiO$_2$) and high-silica (49–52 wt% SiO$_2$) mafic rock groups

## Discussion

The rapid increases of Diff (HFSE) from near zero to positive values during the 2.1–1.8 Ga interval (Fig. 4a) reflect the dramatically increasing contribution from intracontinental alkali basalts to the Earth's mafic rocks. We propose that this abrupt shift represents an enhanced cooling pace of the mantle, as a consequence of the initiation of Phanerozoic-like continuous subduction.

Previous numerical modeling studies suggested that the postulated subduction style of Archean plate tectonics is episodic[47,48] and shallow[3,5,10,49]. The principal reasoning for an episodic subduction style in the Archean is the higher mantle temperature[10,47], which resulted in more extensive partial melting

of the mantle[9]. The higher degree of mantle melting produced a much thicker oceanic crust than today[10,50]. Consequently, as such crustal material is significantly less dense than the lithospheric mantle, the thicker oceanic crust would have reduced the overall density of the bulk oceanic lithosphere[9,47]. At the same time, the higher mantle temperature beneath the oceanic crust certainly would have reduced the strength of the subducting slab and formed a weaker oceanic plate. With the thicker and weaker oceanic crust, a greater tensile stress should exist between the buoyant horizontal oceanic plate and the subducted eclogitized slab[9,47]. In the numerical model provided by van Hunen and van den Berg[47], the Archean subduction model exemplifies a discontinuous subduction behavior, as a slab would frequently break

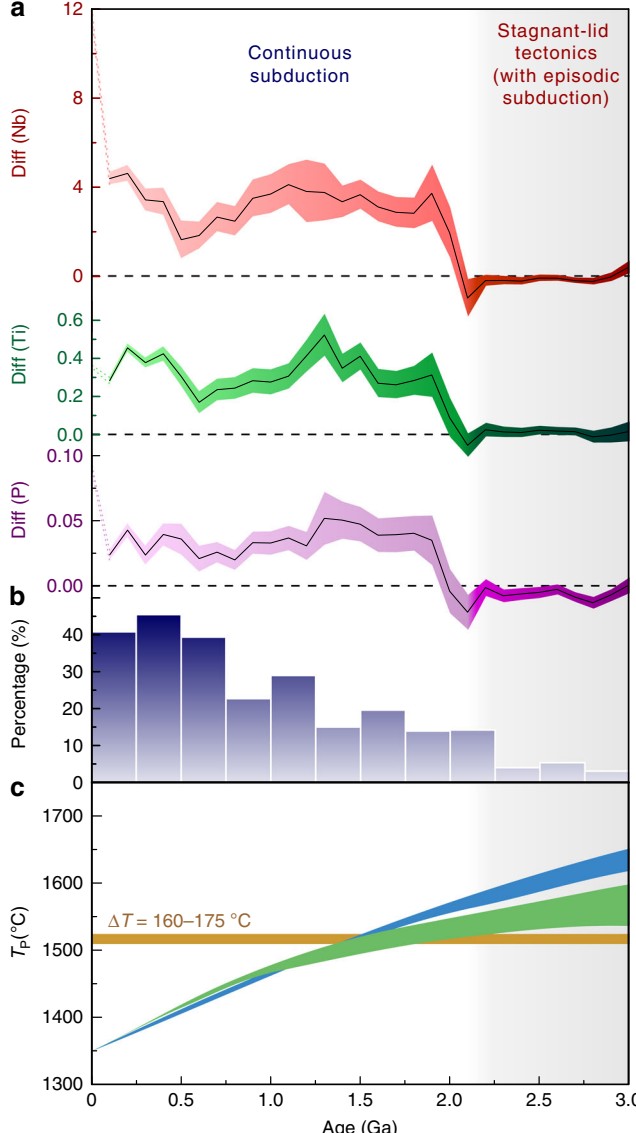

**Fig. 4** The evolution of alkali basaltic magmatism throughout Earth's history. **a** Secular variations of Diff (Nb), Diff (Ti), and Diff (P), respectively, from 3.0 to 0 Ga. The Diff (Nb, Ti or P) is a new geochemical proxy defined in this study to monitor the increase of alkali basaltic rocks. A moving average is applied with the window width equal to 0.5 Ga, while the moving step width is 0.1 Ga. The black solid line in the middle of the three bands from top to bottom are the mean values of Diff (Nb), Diff (Ti), and Diff (P), respectively. The red, green, and purple bands denote standard deviation (s. d.) estimated by the Monte Carlo method. All the Diff (Nb), Diff (Ti), and Diff (P) show rapid increases from nearly zero to positive values at ca. 2.1 Ga. The gray area shows the time range (>2.1 Ga) dominated by stagnant-lid tectonics with episodic subduction, while the white area is the stage (2.1-0 Ga) of continuous plate subduction. The enhancement at the last 0.1 Ga is caused by sampling bias. **b** The proportion (%) of alkali basalts in relation to all mafic rocks at 0.25-Ga time intervals from 3.0 to 0 Ga. The alkali basalt is defined by the Rittmann Serial Index ($\sigma$) for rocks greater than 3.5[62]. $\sigma = (Na_2O + K_2O)^2/(SiO_2-43)$. **c** Thermal evolutions models of the mantle as estimated by Korenaga[13]. Green and blue shadings correspond to the dehydration stiffening hypothesis (closed-system evolution) and the mantle hydration hypothesis (open-system evolution), respectively. The closed-system and open-system models are both end-member scenarios. The orange horizontal bar at $\Delta T = 160$–175 °C (relative to present-day values) shows the threshold of triggering continuous subduction[10]

off from the trailing oceanic plate and sink down into the mantle transition zone. This loss of slab pull may frequently lead to a temporal cessation or slowdown of the oceanic plate subduction. Such an episodic recycling of oceanic crust would not decrease the mantle temperature efficiently. Herzberg et al.[46] proposed that the mantle was warmed to a thermal maximum at ca. 3.0–2.5 Ga (Fig. 4c) based on the observation of secular MgO contents of non-arc basalts, and explained it as internal heating exceeding surface heat loss during the Archean. We suggest that the lack of continuous subduction might have impeded efficient heat loss, which contributed to the mantle warming in the Archean and the subsequent slow decrease of mantle temperature.

Although the thermal history of the Earth is poorly constrained, there are multiple lines of evidence that the mantle temperature during the late Archean (ca. 3.0-2.5 Ga) was ~200–250 °C hotter than its present temperature (i.e. $\Delta T = 200$–250 °C)[13,46,51] and experienced a secular cooling process during the Proterozoic and Phanerozoic[5,13,46]. Numerical modelings suggest that the oceanic plates could underthrust beneath the continental plates along shallow subduction zones when the mantle temperature ($T_P$) was <250 °C above its present temperature[10,49]. However, because the plates were likely weak and deformed internally, the subducting oceanic plate might break-off frequently at shallow depths[7,9,47]. Continuous plate subduction could only be established after the upper mantle became cold enough ($\Delta T = 175$–160 °C) and thus the oceanic plates became mechanically strong. The mantle potential temperature ($T_P$) at 2.1 Ga was around 1520–1580 °C as proposed by Korenage[13], ~170–230 °C higher than the present-day mantle (Fig. 4c). Before 2.1 Ga, the temperature ($T_P$) had exceeded the threshold temperature ($\Delta T = 175$–160 °C) for triggering continuous plate subduction[10]. Once the mantle after 2.1 Ga was cold enough to generate a stronger subducting oceanic slab, the oceanic plate was subducted into the mantle along convergent plate boundary in an uninterrupted succession unless terminated by continental collision. Consequently, the transition of plate subduction style could significantly enhance the mantle cooling by recycling increasing volumes of cold oceanic crust into the mantle continuously[31–33]. In this circumstance, heat loss of the mantle largely exceeded the internal heating from radioactive decay, and therefore rapidly decreased the mantle temperature. The accelerated decrease of mantle temperature resulted in a lower degree of partial melting of the sublithospheric mantle away from the subduction zones, which gave rise to increasing amounts of continental alkali basalts with high Nb, Ti, and P concentrations and relatively low-silica contents. Therefore, we conclude that the rapid increase of Diff (HFSEs) at ca. 2.1 Ga (Fig. 4a) is a consequence of the initiation of continuous subduction and enhanced mantle cooling.

The rapid decrease of the mantle temperature around ca. 2.1 Ga was also noticed by previous studies on the changes in geochemical compositions of komatiites[32,33]. The average MgO contents in komatiites older than 2.4 Ga were >25 wt%, and dropped to around 20–22 wt% in komatiites with ages of 2.2–1.8 Ga (Supplementary Fig. 2). Condie et al.[32,33] proposed that the compositional change in komatiites and a decrease of magma generation temperatures ($T_g$) of basalts derived from depleted mantle between 2.5 and 2.0 Ga reflect an enhanced cooling of the mantle by subduction[32]. To further verify the drop of mantle temperature around ca. 2.1 Ga, we calculated the secular proportions of those mafic rocks generated by high degrees of mantle melting (e.g. high MgO, low Rittmann Index) in relation to all mafic rocks from 3.0 to 0 Ga (Supplementary Fig. 3; Supplementary Tables 7 and 8). The proportion of high-degree melting rocks is large in the stagnant-lid stage and show a dramatic drop around 2.1 Ga, which is most likely a consequence of the

accelerated decrease of mantle temperature caused by the onset of continuous subduction. Although the decrease in degree of mantle melting may be partly due to non-linear melting of the mantle[20], the cooling rate should definitely be enhanced by increasing the volume of cold oceanic crusts subducted into the mantle.

An assortment of evidences of magmatic and orogenic activities indicate a tectonic transition in the early Paleoproterozoic. The 2.4–2.2 time interval includes a minimum of magmatic records, which has been previously recognized as a magmatic shutdown[52] and recently proposed as a tectono-magmatic lull[53] due to the additional evidences of igneous rocks and detrital zircons within this interval[54]. Spencer et al.[53] linked the magmatic flare-up subsequent to this magmatic lull with the dramatic growth of continental crust and the transition from supercraton to supercontinent cycle. Brown and Johnson[3] also noticed a rise in thermal gradients of high-temperature/pressure ($dT/dP > 775\,°C$ $GPa^{-1}$) metamorphism ~2.3–2.1 Ga and the spread of meta-morphic rocks with ages from 2.1 to 1.5 Ga, and interpreted them as the reconfiguration from supercratons to the first super-continent. These perspectives are in accord with our proposal because the ongoing plate convergence of the supercontinent cycle is driven by continuous plate subduction that eventually led to continental collisions and supercontinent assembly[4,55]. The onset of continuous subduction coincided with the start of a ca. 2.1–1.8 Ga global-scale collisional orogeny[49,56], which was followed by the formation of Earth's first large supercontinent, Nuna (also named Columbia)[53,56,57].

Our proposal is also consistent with the formation at ca. 2.0 Ga of the Earth's oldest ophiolites so far discovered[26,27], as well as the recently reported Paleoproterozoic low temperature/pressure ($dT/dP < 375\,°C\,GPa^{-1}$) metamorphic rocks (ca. 2.1–1.8 Ga)[3,28–30]. Although there was no record of ophiolites during the whole Mesoproterozoic, the general paucity of passive margins and the absence of evidence of orogenesis throughout the Mesoproterozoic indicate no significant breakup and reassembly of continental fragments in the transition from Nuna to Rodinia[49], which probably also accounts for the lack of ophiolites. The scarcity of blueschists during the Mesoproterozoic implies that the con-tinuous plate subduction identified in the Paleoproterozoic may be somewhat different from the previously defined modern cold plate subduction, which is characterized by the progressive appearances of blueschists typically in the Neoproterozoic and Phanerozoic[2,3,12]. However, the initiation of continuous subduc-tion and the following collisional orogeny might have contributed to the occasional occurrences of localized cold subduction records and the formations of low temperature/pressure (T/P) meta-morphic rocks at ca. 2.1–1.8 Ga[3,28–30].

In summary, the plate subduction style has changed from the early episodic subduction in the stagnant-lid dominated tectonic regime to the later continuous subduction due to the secular cooling of the mantle. The mantle temperature during the stagnant-lid stage was much higher than its present temperature, thereby accounting for the Archean scarcity of continental alkali basalts due to the higher degree of mantle melting. As the mantle gradually cooled after the Archean, oceanic slab would become strong enough to stabilize a continuous subduction. The mantle cooling was accelerated thereafter due to the continuous recycling of cold oceanic crust. The drop of mantle temperature resulted in a rapidly decreasing degree of mantle melting, which gave rise to increasing amounts of continental alkali basaltic rocks enriched in Nb, Ti, and P. Statistical analysis of igneous geochemical data indicates that the Nb, Ti, and P concentrations in low-silica mafic rocks has increased significantly since ca. 2.1 Ga. The beginning time of continuous subduction is estimated as ca. 2.1 Ga based on this systematic change. This proposal is coincident with the

compositional change of komatiites[32,33], the decrease of the magma generation temperature ($T_g$) of basalts derived from depleted mantle[31,32], the start of ca. 2.1–1.8 Ga global-scale col-lisional orogeny[56], the transition from the supercraton to the supercontinent cycle[3,53], the oldest unequivocal ophiolites[26,27], and the earliest records of low temperature/pressure (T/P) metamorphic rocks[3,28–30].

## Methods

**Data compilation and filtering.** Fortunately, a sufficient quantity of representative rock geochemical data now exists to chronicle the secular evolution of plate tec-tonics over time[58]. Thus, geochemical data from previous studies have here been compiled to perform a statistical analysis on the secular variation of global mag-matic activities and style changes of plate tectonics. A freely accessed rock geo-chemistry database, EarthChem[45], has integrated geochemical data from abundant publications and some other databases (e.g. GEOROC, PetDB, and USGS), which spans the geological history of the Earth from ca. 3.8 Ga to the present. Although the quantity of Precambrian data is much less than for the Phanerozoic, the col-lection of random publications is a relatively balanced process for different ages. Specifically, the database is not expected to arbitrarily generate a sampling bias for rocks of a certain age. As such, we considered that the geochemical data of igneous rocks from the EarthChem database are representative of global magmatism throughout Earth's history.

In this study, we downloaded all the geochemical data for the igneous rocks from the EarthChem database, including the whole-rock major and trace element concentrations, ages, and geographic coordinates. The rock samples come from different continents and island arcs around the world. Samples lacking their $SiO_2$ concentration were omitted. Lithologies of rocks in the downloaded dataset encompass mafic (45–52 wt% $SiO_2$), intermediate (52–63 wt% $SiO_2$), and felsic rocks (>63 wt% $SiO_2$).

In consideration of their distinct geochemical characteristics, we used only arc and continental mafic rocks (45–52 wt% $SiO_2$) in this study. Rocks from the oceanic plateau, floor, and islands were excluded from our dataset to balance these geological settings with oceanic rocks of Precambrian ages, which are scarcely preserved due to plate subduction (Supplementary Fig. 4). Rocks older than 3.5 Ga were not included either, because the plate tectonics regime before 3.5 Ga is still controversial[14,20]. Rocks in some subduction-unrelated tectonic settings (e.g. continental flood basalts) are included because the proportion of subduction-related to subduction-unrelated rocks is associated with the thermal conditions and plate tectonic styles. Considering that komatiites, lamprophyres, and carbonatite often interest geologists and geochemists and might have been over sampled, we checked the original publications and removed those rock types to avoid the sampling bias. Besides, we also remove the dolerite/diabase dykes from the dataset because most of those dolerite/diabase dykes have undergo significant crustal contamination, which cause the additional gain of incompatible elements[59,60]. A final dataset containing 55,107 whole-rock geochemical data of mafic igneous rocks (Supplementary Fig. 5; Supplementary Data 1) covering 3.5-0 Ga was prepared for the statistical analysis.

We extracted two subsets of igneous rocks with ages of 5 Ma or less from the Southern Andes of South America and East Asia to characterize the geochemical behaviors of elements across subduction zones. The reason to choose such young rocks is that their current locations most closely represents the geological settings where they formed. The rocks were also subdivided into arc and intracontinental rock groups based on their locations. For instance, in East Asia, rocks from the Kamchatka Peninsula, Kuril Islands, Japanese Islands, Ryukyu Islands, and Izu-Bonin-Mariana Arc were classified into the arc group, while rocks from the eastern China were classified into the continental group.

**Statistical analysis.** The mean element concentrations corresponding to $SiO_2$ from 45–52 wt% were calculated using the Bootstrap resampling method based on MATLAB (R2014a) encoding. We applied a central moving average smoothing to determine the means. The width of the sample window is set at 2 wt% $SiO_2$, while the step width is set at 1 wt% $SiO_2$. We executed 10,000 iterations for the Bootstrap resampled subsets. Mean value and corresponding standard error for every 1 wt% $SiO_2$ were obtained by calculating the average value and standard deviation of those 10,000 resampled subsets. Diagrams of the combined Nb-$SiO_2$, $TiO_2$-$SiO_2$, and P-$SiO_2$ compositional trends are shown in Figs. 1 and 2, respectively.

We noticed that the mean HFSE and P concentrations in low-silica mafic rocks (45–49 wt% $SiO_2$) in the Phanerozoic are much higher than those in high-silica mafic rocks (49–52 wt% $SiO_2$), whereas the mean HFSE and P concentrations are slightly lower in low-silica mafic rocks in the Archean. Hence, a new geochemical proxy Diff (HFSE) to monitor the HFSE-$SiO_2$ (or P-$SiO_2$) compositional trend of mafic rocks is defined by the following formula:

$$\text{Diff(HFSE)} = \omega(\text{HFSE})_{low-Si} - \omega(\text{HFSE})_{high-Si} \qquad (1)$$

In Formula (1), the $\omega(\text{HFSE})_{low-Si}$ is the mean HFSE (or P) in rocks with 45–49 wt% $SiO_2$, while $\omega(\text{HFSE})_{high-Si}$ is the mean HFSE (or P) in rocks with 49–52 wt % $SiO_2$.

The Diff (HFSE) over time (from 3.0 to 0 Ga) are calculated based on MATLAB encoding. The Diff (HFSE) of every 0.2 Ga time bins was obtained (Supplementary Fig. 6; Supplementary Tables 9–11). Because there are a limited number of samples for certain time intervals, we use the moving average method to calculate the value for a specific age. The width of the sample window is set at 0.5 Ga to include an adequate number of samples for the calculation, while the moving step width is set at 0.1 Ga. For instance, the Diff (Nb) for 1.5 Ga is calculated by the rocks formed between 1.5 and 2.0 Ga, while the Diff (Nb) for 1.4 Ga is calculated by the rocks formed between 1.4 and 1.9 Ga. We set the lower boundary of a sample window as the age of this window because what we are concerned with is the variation owing to newly involved younger samples. The means and standard errors of $\omega(HFSE)_{low-Si}$ and $\omega(HFSE)_{high-Si}$ are calculated by Bootstrap resampling based on the MATLAB (R2014a) function. The Diff (HFSE) and corresponding standard deviation are estimated by Monte Carlo simulation. According to the Central Limit Theorem, regardless of the distribution of a dataset, the means of randomly sampled subsets will naturally tend towards a Gaussian distribution. We created two synthetic Gaussian-distributed datasets for the means of $\omega(HFSE)_{low-Si}$ and $\omega(HFSE)_{high-Si}$ for every 0.1 Ga. The mean and standard deviation of a synthetic dataset was set at the mean and corresponding standard error of $\omega(HFSE)_{low-Si}$ or $\omega(HFSE)_{high-Si}$, respectively. Each synthetic dataset included 10,000 simulated data, for which 10,000 simulated Diff (HFSE)s were obtained through use of formula (1). Finally, the means and standard deviations ($1\sigma$) of the Diff (HFSE)s were estimated for all the ages from 3.0 to 0 Ga, calculated at 0.1 Ga intervals.

## Data availability

The datasets used in this study are provided as Supplementary Data 1. The source data underlying Figs. 1 and 2, and Supplementary Fig. 2 are provided as a Source Data file. The Supplementary Data 1 has been deposited in GitHub repository (https://github.com/CodrIocas/Subduction).

## Code availability

The computer codes used in this study are provided as supplementary information. All the relevant data and codes are freely available in GitHub at https://github.com/CodrIocas/Subduction. The MATLAB codes are provided under a GNU GPL v2.0 open-source license.

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

## Acknowledgements

C. Langmuir and A. W. Hofmann provided constructive suggestions for the texts and the statistical algorithm used in this paper. This study was supported by Strategic Priority Research Program of the Chinese Academy of Sciences, Grant No. XDB18020102, MOST of China 2016YFC0600408 and NSFC 41473029. H.L. was supported by an IOCAS Postdoctoral Fellowship.

## Author contributions

H.L. conducted the MATLAB coding and interpreted the results. W.D.S. contribute to the interpretation of results. H.L. wrote the manuscript with suggestions from all co-authors. M.T. improved the structure of the manuscript. R.Z. improved the English and contributed to the scientific presentation. All authors participated in further discussions.

## Additional information

**Competing interests:** The authors declare no competing interests.

