## [Peer Review File · Nature Communications]

Reviewers' comments:

Reviewer #1 (Remarks to the Author):

This paper makes an interesting observation from extraction of data from an internet database about a change in chemical compositions through time. To put it simply, it is the observation that there are few alkalic basalts prior to two billion years ago. Alkalic rocks today have high Nb, Ti and P contents at low Si contents, and arc basalts have higher Si and lower Nb, P and Ti, so if one plots such incompatible elements vs. Si there is a negative slope in rocks today, and not in older rocks. The authors then try to see how this has changed with time by turning this slope into a simple index and plotting it vs. time, with the interesting result shown in their Figure 4. They then interpret this change in terms of models of slab subduction, previously published by others, to argue for a transition from "hot subduction" to "cold subduction" 2.1 billion years ago.

The authors are to be commended for trying to quantify how the proportions of alkaline rocks have changed with time, and the results they show are indeed interesting. In my view, however, their work has just begun and this paper is premature. First, there is a worry about the data. (a) There is fifty times as much data for Phanerozoic rocks as Archean rocks. For some silica contents there are only a few hundred data points, and there is a sampling bias because everyone likes to analyze komatiites. Alkali basalts are usually little cinder cones. They would be hard to preserve, and rare to sample. A factor of fifty.... It's a lot to consider. (b) In supplementary table 4 there is no indication of the data density. If there are only a few hundred data points for the entire Archean, presumably some of these hundred million year bins have only some tens of data points?? I would suggest publishing the actual variation diagrams for each age interval. It is not clear to me how robust the time series is, no matter whether boot strapping was done or not. But probably the overall trends in Figure 4 are about right—because there are low Si high degree melts in the Archean, which will have low incompatible elements, and low Si low degree melts in the Phanerozoic, which have high incompatible elements. That is pretty straightforward, and I think most people would expect that from common sense. What would be novel would be the time series aspect, shown in Figure 4. Let's accept that as real. Then what is the significance of this observation?

The authors refer to recent thermal models to try to relate this result to variations in the style of subduction over Earth's history. That is one interesting possibility, but there is so much more to consider. They need to really think through all the various hypotheses that might account for the data, and evaluate them rigorously. Yes, variations in mantle temperature with time ought to have had an influence on subduction, and many have proposed that it has, but are the observations in this paper really a telltale sign of change in subduction? Do the data really provide a new time constraint on temporal change in slab behavior? Is that really the only possible explanation for the observations, and therefore a definitive result about the timing of thermal evolution of subduction zones? Maybe, maybe not. Let's consider some of the complexities.

(1) The authors propose that all modern subduction has slab rollback as a consequence of cold subduction. But not all modern convergent margins have slab rollback. There is rollback in the western Pacific, and back-arc basins showing extension. But there is no rollback and no back-arc basins in the eastern Pacific. That is because the Atlantic is opening, causing N. America and S. America to move westwards, leading to compressional rather than extensional convergent margin regimes. Not all slabs today lead to back-arc extension.

(2) The authors assume there is one slab temperature at any particular time, but thermal models of modern slabs show probable variations of hundreds of degrees between young, slowly subducting slabs and hot, quickly subducting slabs. New slab thermometers such as H₂O/Ce had led to proposed variations of perhaps three hundred degrees in modern slabs! So, this is a good test. Are there no alkali basalts behind hot slabs today, and lots of alkali basalts behind cold slabs? No, hot slabs in Mexico and the Cascades have lots of alkalic back-arc volcanism, and some colder slabs have very little (e.g. the Aleutians, the Lesser Antilles).

(3) The authors propose that alkalic magmatism is the result of back-arc extension due to slab rollback. But there is alkalic volcanism today in many settings—ocean islands, intra-continental rifts that have nothing to do with slabs (e.g. E. Africa, France, the Rhine Graben) and the widespread extension in the western United States that is not due to slab rollback but possibly the subduction of a spreading center, which may also be the case in the southern Andes. So the connection is not between slab temperature, rifting and alkalic volcanism; alkalic volcanism today occurs in every tectonic setting, even on the volcanic front of many modern arcs. Grenada in the Lesser Antilles, shoshonites in western Mexico, etc. Yes, in a portion of the western Pacific there is slab rollback and back-arc alkalic volcanism, but you get alkalic volcanism everywhere, most of it not associated with slab rollback. The connection between slab temperature and alkalic volcanism is not established in the modern Earth, let alone the ancient Earth where we do not know the tectonic setting of the samples.

(4) Alkali basalts are generally considered to be low degree mantle melts. Small extents of melting beneath thick lithosphere have an almost universal association with modern alkalic volcanism—I suspect a much more universal observation than a subduction connection. So maybe the change in alkalic volcanism through time is just a consequence of decreasing mantle temperature. In ancient times the mantle was hotter, so Earth did not produce low degree mantle melts, because the temperature was higher, and the lithosphere likely thinner. Isn't that perfectly consistent with Figure 3? Low incompatible elements contents across the range of SiO₂ in the Archean—that's right, high mantle temperatures, high extents of melting. Today you get you SiO₂ from deep, low degree melts; in the Archean you got low SiO₂ from high degree melts. Wouldn't that produce the observations? That would have nothing to do with subduction style.

(5) The authors draw an arbitrary line in Figure 4 to try to indicate a key boundary in Earth's history. But if they are going to make a lot out of this index, wouldn't they need to explain the very large spike in the data between 1 and 1.5?. And then why the steep decline between 1.2 and 0.6 billion years? And why the steep rise in the last fifty million years or so? The magnitude of these changes is as large as the long term trend they emphasize, but no explanation or consistent model is given to account for most of the variation observed.

In sum, the authors have an interesting observation here, and a commendable attempt to see how proportions of mafic rocks vary through time. They are just at the beginning of exploring the meaning of these observations. This paper is not ready yet for publication in any journal until it is reframed, perhaps something like this: Here is the interesting observation from database mining—their Figure 4. How can we understand this variation? Let's look at the modern Earth and see where alkalic volcanism occurs. Is there an association with subduction? With cold slabs and not hot slabs? What can we say from modern observations about the cause. This leads to models based on well constrained tectonic settings and thousands of data points. That would then lead to many possible explanations for temporal change—multiple hypotheses. How might we distinguish among them?

.... On present evidence, I doubt a clear distinction can be made. There is a change in subduction over Earth history, and an absence of alkali basalts in early Earth history, but does the change really mark a global change in the character of subduction? Maybe it is just another manifestation of the change in mantle temperature, independent of subduction. A more thoughtful and thorough consideration could lead to a fine paper on the basis of the database mining. As it stands, this paper simply does not have a convincing case to be made.

Reviewer #2 (Remarks to the Author):

Liu and coauthors present a statistical analysis of a large dataset of whole-rock geochemistry of mafic samples spanning some 3.5 Gyr.

The authors propose two specific endmember modes of "hot" (perhaps episodic) vs "cold" (modern) subduction, and propose a transition from the former to the latter has occurred over the past 3 Gyr. The proposed evidence for this shift is an increase in the abundance of alkali basalts (identified on the basis of high HFSE and P contents) over the past 3 Gyr.

In particular, the authors propose that alkali basalts act as a tracer of continental rifting driven specifically by slab rollback. They further claim that hot subduction cannot sustain slab rollback, and that this in turn explains the "scarcity of continental alkali basalts" in the geologic past.

While a transition from "hot" to "cold" subduction over the past 3 Gyr is broadly plausible (perhaps even tautologically inevitable) given secular mantle cooling, there is at least one, rather glaring, problem with the specific narrative presented here: of the classic continental rifts active on Earth today (East African, Rio Grande, and Baikal), not a single one is driven by, or even significantly associated with, slab rollback.

While the authors highlight several other localities where alkali basaltic magmatism may occur in a back-arc setting, the line from slab rollback to alkali basaltic magmatism is strained and nonunique; any type of low-degree mantle decompression melting will suffice to produce alkali basalts. Further, while we may expect some alkali basaltic magmatism in back arcs, back-arc magmatism typically displays a mixing between decompression and flux melting signatures, thus complicating the high HFSE enrichment expected for pure decompression-derived alkali basalts by potentially mixing in an arc HFSE depletion signature. The use of high HFSE as a proxy for slab rollback is thus rather bizarre.

Concerningly, the criteria used as characteristics of continental alkali basalts (high HFSE and P contents, along with low silica) are more specifically signatures of low-degree mantle decompression melting, regardless of the tectonic setting. Consequently, we ought to observe an increase in alkali basaltic magmatism as a direct consequence of secular mantle cooling, independent of tectonic style.

A more direct interpretation of the dataset presented here would be that it confirms our expectation that mantle potential temperature and melting extent have declined over the past 3 Gyr – which, though noteworthy, may not yield the sort of novelty or impact factor the authors are apparently hoping for.

Review of Liu et al.: “Hot-cold plate subduction transition marked by the rise of alkali magmatism ~2.1 billion years ago”

I really enjoyed reading this paper, which presents statistical evidence in support of the widespread development of continental alkali basalts from 2.1–1.8 Ga. The authors provide convincing evidence that these types of rocks form in backarcs on the modern Earth, and link their widespread development in the Paleoproterozoic to continuous (or at least long-lived) subduction rollback that was likely made possible due to secular cooling of the mantle. The authors conclude that ~2.1 Ga marks the commencement of ‘modern-style cold subduction’.

The paper is generally well written and presents a smart but simple way of thinking about the data. It deals with exactly the sorts of things that interest me, and which are clearly of broad interest to the wider community. How, why and when plate tectonics became the dominant geodynamic ‘mode’ is a hot topic. I think this paper provides some strong support of a major change in the Paleoproterozoic, but I am not convinced that it is the change to ‘modern-style cold subduction’, at least not as far as I (and I would content many others) understand the term.

The main problem with the paper is a complete disregard of anything other than ‘hot subduction’ as the main tectonic mode on Earth between 2.1 and 3.5 Ga. Others have made arguments for a dominance of stagnant lid tectonics for much or perhaps all of the Archaean that gets not a single mention (see, for example, Cawood et al 2018 for some of the evidence: Cawood, P.A., Hawkesworth, C.J., Pisarevsky, S.A., Dhuime, B., Capitanio, F.A. and Nebel, O., 2018. Geological archive of the onset of plate tectonics. *Philosophical Transactions of the Royal Society A: Mathematical, Physical and Engineering Sciences*, 376).

As noted recently by Brown and Johnson (Secular change in metamorphism and the onset of global plate tectonics. *American Mineralogist*, 103, 181–196, 2018.), which uses a much larger metamorphic database than Brown (2006), and might warrant a mention, the ‘modern-style’ of subduction, representing cold collision and deep subduction of the slab prior to breakoff, is associated with the formation of low dT/dP rocks, most conspicuously blueschists. While there are rare examples of such rocks as old as ~2.1 Ga, as the authors note, they do not appear widely until the Neoproterozoic (~0.75 Ga). This is starkly illustrated below, using an updated database comprising 564 data points from a paper (Brown, M., & Johnson, T. (2019). Metamorphism and the evolution of subduction on Earth. *American Mineralogist*, in review) that I will try to remember to attach.

Of course, the authors might argue that this is a preservation issue, and that the much older (Paleoproterozoic) data are the only remaining examples of low T/P rocks that were much more widespread. However, it seems more likely to me that they record localised examples of cold subduction, not (global) plate tectonics. In my view, arguing for the onset of global 'modern' subduction would require much more support than here. In any case this needs much more discussion (see also below).

What the new metamorphic data do show is the emergence of 'paired metamorphism' sometime in the late Archean to Paleoproterozoic. The Neoproterozoic metamorphic data of Brown & Johnson (2018, and in review) can be described by a unimodal Gaussian distribution. Sometime in the Paleoproterozoic, the data exhibit a distinctly (and statistically significant) bimodal character that becomes increasingly bimodal prior to the widespread appearance of blueschist that records the onset of (what we would argue is) 'modern' plate tectonics--see density plot below. This is something else we have currently in review (Holder et al.).

To summarize, I think the authors are identifying the onset of continuous/long-lived subduction and rollback at 2.1 Ga (i.e., the onset of plate tectonics, a globally-linked network of narrow plate boundaries) from an earlier regime dominated by stagnant lid tectonics and/or shallow/failed/hot subduction. I do not think ~2.1 Ga it is the establishment of the modern regime, which is characterized by low T/P metamorphism and the widespread formation of blueschists. Again, the authors are under no obligation to agree, but they do need to discuss these possibilities (that are already published).

All in all, I think this is a strong paper with some significant findings. However, I would personally rethink the implications of the data presented here within the context of the existing metamorphic and other data. The paper could be expanded by a quarter to a third to discuss several aspects in better details. I believe Nature Comms permits longer (>4 pages) format papers, but the appropriate link was broken when I tried it:

http://www.nature.com/ncomms/about_journal.html

Some specific comments keyed by line number follow:

Lines 57–60: Yes, but what is the evidence that these examples record global rather than local episodes?

73–74: This sentence is a bit clumsy and could be redrafted.

125–128: I disagree that most previous studies agree what is stated. To reiterate, there is no absolutely no mention of ‘stagnant lid’ (thick plateaux/‘vertical’ tectonic/sagduction/etc.) models that many believe are perfectly plausible. This is a major shortcoming of the paper.

131: I don’t understand the meaning of ‘differentiation trends’. Unless I’m mistaken, the compositional signatures you are describing are a function of the source not of magmatic fractionation. Is ‘secular trends’ or ‘compositional trends’ better?

142–144: Or they reflect a change from a pre-plate tectonic regime to a plate tectonic regime. No problem if you disagree, but the evidence needs to be discussed.

159–152: Definitely correct. Might this warrant a reference or two?

185–186: There is a clear change here, but there are others (notably the big peak at ~1.2 Ga and the low at 0.6 Ga). I assume these are interpreted in terms of the supercontinent cycle, but some more details would be good...

201–203: ...as the variations do not seem to have the same episodicity as the metamorphic or detrital (and other) zircon record.

277–278: To reiterate, I have big problems with the assumption that ‘hot subduction’ was the main or only game in town from 3.5 to 2.1 Ga. Some regard needs to be paid to alternative models.

Fig. 4. I may have missed it, but some details as to how the curves in (a) were calculated would be helpful. I assume they are a moving mean, but for what window size and what step?

A couple of refs for ‘sagduction’ models:

Bédard, J.H., 2006. A catalytic delamination-driven model for coupled genesis of Archaean crust and sub-continental lithospheric mantle. *Geochimica et Cosmochimica Acta*, 70(5), pp.1188-1214.

Johnson, T.E., Brown, M., Kaus, B.J. and VanTongeren, J.A., 2014. Delamination and recycling of Archaean crust caused by gravitational instabilities. *Nature Geoscience*, 7(1), p.47.

A point-by-point response

Here is our point-by-point response to the reviewers' comments.

Reviewer #1 (Remarks to the Author):

This paper makes an interesting observation from extraction of data from an internet database about a change in chemical compositions through time. To put it simply, it is the observation that there are few alkalic basalts prior to two billion years ago. Alkalic rocks today have high Nb Ti and P contents at low Si contents, and arc basalts have higher Si and lower Nb, P and Ti, so if one plots such incompatible elements vs. Si there is a negative slope in rocks today, and not in older rocks. The authors then try to see how this has changed with time by turning this slope into a simple index and plotting it vs. time, with the interesting result shown in their Figure 4. They then interpret this change in terms of models of slab subduction, previously published by others, to argue for a transition from "hot subduction" to "cold subduction" 2.1 billion years ago.

The authors are to be commended for trying to quantify how the proportions of alkaline rocks have changed with time, and the results they show are indeed interesting. In my view, however, their work has just begun and this paper is premature. First, there is a worry about the data. (a) There is fifty times as much data for Phanerozoic rocks as Archean rocks. For some silica contents there are only a few hundred data points, and there is a sampling bias because everyone likes to analyze komatiites. Alkali basalts are usually little cinder cones. They would be hard to preserve, and rare to sample. A factor of fifty.... It's a lot to consider.

R: As pointed out by the Reviewer, we found a lot of komatiites in the dataset. Therefore, we removed all komatiites and rocks with MgO greater than 18 wt%. We also check the publications of the high-MgO rocks and removed all komatiitic

basalts. Thanks for this comment.

Although the sample number of the Phanerozoic is much larger than that of the Proterozoic and Archean, both the alkali and sub-alkali mafic rocks in the Phanerozoic are more abundant than those in the Proterozoic and Archean. As such, the proportion of alkali mafic rocks vary in a reasonable range (Figure 4b). For instance, the sample number in the 0.5-0.25 Ga time interval (3667 samples) is five times larger than the sample number in the 0.75-0.5 Ga interval (746 samples), but the proportions of alkali basalts in both intervals are roughly equal (Figure 4b). Therefore, the preservation and sampling problems for alkali and sub-alkali mafic rocks are the same. When we calculate the average Nb, Ti and P concentrations of low-silica and high-silica mafic rocks for a specific time interval, the results will be dependent of the proportion of alkali mafic rocks but independent of sample numbers. One exception is in the latest 0.1 Ga. The alkali mafic rocks with ages of 50-0 Ma comprise too many samples from continental rifts (especially from the East African Rift) and volcanic cones in the broad areas of continents (Supplementary Figure 3a), while the alkali mafic rocks in previous times include more rocks formed by melting of mantle plumes and less samples from the continental extensional settings (Supplementary Figure 3b). Alkali basalts from continental extensional settings contain more Nb and P but variable Ti (Farmer, 2014. Treatise in Geochemistry), accounting for the high Diff (Nb) and Diff (P) but moderate Diff (Ti) for the latest 0.1 Ga. In our dataset, the sample number in the Proterozoic is generally equal to that of the Archean. However, the proportion of alkali mafic rocks in the Archean is less than 5%, whereas the proportion in the Mesoproterozoic varies in the range of 15-30%, with an average of 22%. Thus, the observed increasing magnitude of alkali basaltic magmatism at ca. 2.1 Ga is reliable.

(b) In supplementary table 4 there is no indication of the data density. If there are only a few hundred data points for the entire Archean, presumably some of these hundred million year bins have only some tens of data points?? I would suggest

publishing the actual variation diagrams for each age interval. It is not clear to me how robust the time series is, no matter whether boot strapping was done or not. But probably the overall trends in Figure 4 are about right—because there are low Si high degree melts in the Archean, which will have low incompatible elements, and low Si low degree melts in the Phanerozoic, which have high incompatible elements. That is pretty straightforward, and I think most people would expect that from common sense. What would be novel would be the time series aspect, shown in Figure 4. Let's accept that as real. Then what is the significance of this observation?

R: The data density is provided in the revised Supplementary Table 4-6. We agree with the reviewer's suggestion of publishing the actual variation diagrams for each age interval. We calculated the Diff (Nb), Diff (Ti) and Diff (P) for each 0.2 Ga age interval from 3.0 to 0 Ga without applying moving average smoothing (see Supplementary Figure 6). The trends of Diff (HFSE) in Supplementary Figure 6 are very similar to those in Figure 4a. Diff (Ti) and Diff (P) show minus values at 2.4 Ga due to the lack of sufficient samples in the 2.5-2.3 Ga interval. The moving average can fix this problem (Figure 4a). Diff (Nb) and Diff (P) show very high values at 0.1-0 Ga due to the sampling bias, which has been explained in the above section and the figure captions of Supplementary Figure 3 and 6.

As the reviewer commented, the overall trends in Figure 4a are caused by decreasing degrees of mantle melting from the Archean to the present. We highly appreciate this proposal, which helped us understand the implication of the rise of alkali basaltic rocks. The sharp increase of Diff (HFSE) at ca. 2.1 Ga reflect a rapidly increasing amounts of alkali basaltic rocks, which is likely to be caused by an enhanced mantle cooling event rather than a gradual cooling process (also see Condie et al. 2016. *Geoscience Frontiers*). We suggest that this enhanced mantle cooling at ca. 2.1 Ga is attributed to the initiation of continuous plate subduction, because the continuous subduction could recycle increasing volume of cold oceanic crusts into the mantle. In contrast, the episodic subduction style before 2.1 Ga might yield frequent slab break-off during the subduction process due to the

weak strength of oceanic plate in a hotter Earth. The slab break-off would lead to a temporal cessation or slowdown of the oceanic plate subduction, as revealed by numerical modelling (van Hunen et al., 2008. *Lithos*; Moyen and van Hunen, 2012. *Geology*). Such an episodic subduction could not decrease the mantle temperature efficiently. Therefore, the transition from episodic to continuous subduction would dramatically accelerate the cooling pace.

The authors refer to recent thermal models to try to relate this result to variations in the style of subduction over Earth's history. That is one interesting possibility, but there is so much more to consider. They need to really think through all the various hypotheses that might account for the data, and evaluate them rigorously. Yes, variations in mantle temperature with time ought to have had an influence on subduction, and many have proposed that it has, but are the observations in this paper really a telltale sign of change in subduction? Do the data really provide a new time constraint on temporal change in slab behavior? Is that really the only possible explanation for the observations, and therefore a definitive result about the timing of thermal evolution of subduction zones? Maybe, maybe not. Let's consider some of the complexities.

R: Agreed. The variations in mantle temperature with time ought to have had an influence on subduction. In addition to the above response, we also integrate many other geological evidences published in previous studies to demonstrate the change in subduction style at around 2.1 Ga. First, it was suggested that the mantle temperature before 2.1 Ga had been >175 °C higher than its present temperature (Korenage, 2013. *Annual Review of Earth and Planetary Sciences*), exceeding the threshold of maintaining a continuous subduction (Sizova et al. 2010. *Lithos*; Brown, 2014. *Geoscience Frontiers*). Second, the rapid drop of mantle temperature at ca. 2.1 Ga (or 2.2-2.1 Ga) has been observed by some other researches. Condie et al. (2000. *American Journal of Science*; 2016, *Geoscience Frontiers*) observed a MgO drop in komatiites and a decrease of magma generation temperatures (T_g) of basalts derived from depleted mantle during the 2.5-2.0 Ga period and interpreted

them as an enhanced cooling of the mantle by subduction. Third, Spencer et al. (2018. *Nature Geoscience*) linked the magmatic flare-up subsequent to the magmatic lull (2.4-2.2 Ga) with the dramatic growth of continental crust and the transition from supercraton to supercontinent cycle. The supercontinent cycle is driven by continuous plate convergence that eventually lead to continental collisions and supercontinent assembly. The onset of continuous subduction exactly coincided with the start of a global-scale ca. 2.1-1.8 Ga collisional orogeny (Zhao et al., 2004. *Earth-Science Review*), followed by the formation of Nuna supercontinent. Fourth, the earliest ophiolites and low T/P ($dT/dP < 375 \text{ }^\circ\text{C/GPa}$) metamorphic rocks formed during 2.2-1.8 Ga. Although there is a sparsity of these records during the whole Mesoproterozoic, Cawood and Hawkesworth (2014. *Geology*) have suggested that the general paucity of passive margins through the Mesoproterozoic and the absence of any evidence for orogenesis indicate no significant breakup and reassembly of these continental fragments in the transition from Nuna to Rodinia. In addition, we calculated the proportion of rocks formed by high degree of mantle melting (high MgO, low Rittmann Index) through time and also noticed a rapid decrease of melting degree at ca. 2.1 Ga (Supplementary Figure 5). Considering the above geological evidences, we suggest that the rise of alkali basaltic rocks at ca. 2.1 Ga reflected an enhanced cooling of the mantle, as a consequence of the onset of continuous plate subduction.

(1) The authors propose that all modern subduction has slab rollback as a consequence of cold subduction. But not all modern convergent margins have slab rollback. There is rollback in the western pacific, and back-arc basins showing extension. But there is no rollback and no back-arc basins in the eastern Pacific. That is because the Atlantic is opening, causing N. America and S. America to move westwards, leading to compressional rather than extensional convergent margin regimes. Not all slabs today lead to back-arc extension.

R: Thanks for these examples. We understood that the relationship between back-arc extension and alkali basalts may be only applicable in the western Pacific. Thus,

we have made a major revision to the manuscript.

(2) The authors assume there is one slab temperature at any particular time, but thermal models of modern slabs show probable variations of hundreds of degrees between young, slowly subducting slabs and hot, quickly subducting slabs. New slab thermometers such as H₂O/Ce had led to proposed variations of perhaps three hundred degrees in modern slabs! So, this is a good test. Are there no alkali basalts behind hot slabs today, and lots of alkali basalts behind cold slabs? No, hot slabs in Mexico and the Cascades have lots of alkalic back-arc volcanism, and some colder slabs have very little (e.g. the Aleutians, the Lesser Antilles).

R: Thanks very much for these examples and comments. We agree with the reviewer. We now realize that the formation of alkali basaltic magma is associated with the thermal state of the mantle rather than the temperatures of the slabs. Now, we emphasize that the mantle temperature before the onset of continuous subduction was high, accounting for the scarcity of alkali basaltic rocks. Whatever a hot or cold slab was subducted on the modern Earth, the subduction of oceanic plate is continuous, even though the break-off happens at depth occasionally. In contrast, slab break-off in the episodic subduction regime in a hotter Earth are very frequent and often occurred at shallow depth, which lead to a temporal cessation or slowdown of subduction (van Hunen et al., 2008. *Lithos*; Moyen and van Hunen, 2012. *Geology*).

(3) The authors propose that alkalic magmatism is the result of back-arc extension due to slab rollback. But there is alkalic volcanism today in many settings—ocean islands, intra-continental rifts that have nothing to do with slabs (e.g. E. Africa, France, the Rhine Graben) and the widespread extension in the western United States that is not due to slab rollback but possibly the subduction of a spreading center, which may also be the case in the southern Andes. So the connection is not between slab temperature, rifting and alkalic volcanism; alkalic volcanism today occurs in every tectonic setting, even on the volcanic front of

many modern arcs. Grenada in the Lesser Antilles, shoshonites in western Mexico, etc. Yes, in a portion of the western Pacific there is slab rollback and back-arc alkalic volcanism, but you get alkalic volcanism everywhere, most of it not associated with slab rollback. The connection between slab temperature and alkalic volcanism is not established in the modern Earth, let alone the ancient Earth where we do not know the tectonic setting of the samples.

R: Thanks very much for the information and comments. We now understand that the link between the eruption of alkali basalts and back-arc extension may be only suitable (or also controversial) in the western Pacific. Thus, we revised the manuscript.

(4) Alkali basalts are generally considered to be low degree mantle melts. Small extents of melting beneath thick lithosphere have an almost universal association with modern alkalic volcanism—I suspect a much more universal observation than a subduction connection. So maybe the change in alkalic volcanism through time is just a consequence of decreasing mantle temperature. In ancient times the mantle was hotter, so Earth did not produce low degree mantle melts, because the temperature was higher, and the lithosphere likely thinner. Isn't that perfectly consistent with Figure 3? Low incompatible elements contents across the range of SiO₂ in the Archean—that's right, high mantle temperatures, high extents of melting. Today you get you SiO₂ from deep, low degree melts; in the Archean you got low SiO₂ from high degree melts. Wouldn't that produce the observations? That would have nothing to do with subduction style.

R: We highly appreciate that the reviewer presented the detailed interpretation of the connection between alkali magmatism and mantle temperature. Through further consideration on the mantle temperature change and more investigations on literatures related to the Paleoproterozoic tectonic change, we propose that the transition of plate subduction style from episodic to continuous subduction resulted in an accelerated mantle cooling and the increasing amounts of alkali basalts at ca. 2.1 Ga. Please see above responses and the revised manuscript.

(5) The authors draw an arbitrary line in Figure 4 to try to indicate a key boundary in Earth's history. But if they are going to make a lot out of this index, wouldn't they need to explain the very large spike in the data between 1 and 1.5?. And then why the steep decline between 1.2 and 0.6 billion years? And why the steep rise in the last fifty million years or so? The magnitude of these changes is as large as the long term trend they emphasize, but no explanation or consistent model is given to account for most of the variation observed.

R: After checking the spike and decline, we found that those changes are caused by the large involvements of dolerite/diabase dyke samples. Most of those dolerite/diabase dykes have undergone variable degrees of crustal contamination (Li et al., 2010. *Precambrian Research*; Zhu et al., 2008. *Lithos*), of which the incompatible elements cannot reflect the degree of mantle melting. Therefore, we removed all the dolerite/diabase dyke samples from the dataset. Now, there is no big change between 1.8-0.1 Ga (Figure 4a). Although it seems there is a low-value range of Diff (Nb) and Diff (Ti) between 0.9 and 0.5 Ga, the values are still above zero and the Diff (Nb), Diff (Ti) and Diff (P) do not vary together. Thus, we think it is not worthy of discussion. The Diff (Nb) and Diff (P) are largely increased for the latest 0.1 Ga, which are caused by sampling bias and has been explained above and in Supplementary Figure 3.

In sum, the authors have an interesting observation here, and a commendable attempt to see how proportions of mafic rocks vary through time. They are just at the beginning of exploring the meaning of these observations. This paper is not ready yet for publication in any journal until it is reframed, perhaps something like this: Here is the interesting observation from database mining—their Figure 4. How can we understand this variation? Let's look at the modern Earth and see where alkalic volcanism occurs. Is there an association with subduction? With cold slabs and not hot slabs? What can we say from modern observations about the cause. This leads to models based on well constrained tectonic settings and

thousands of data points. That would then lead to many possible explanations for temporal change—multiple hypotheses. How might we distinguish among them? ... On present evidence, I doubt a clear distinction can be made. There is a change in subduction over Earth history, and an absence of alkali basalts in early Earth history, but does the change really mark a global change in the character of subduction? Maybe it is just another manifestation of the change in mantle temperature, independent of subduction. A more thoughtful and thorough consideration could lead to a fine paper on the basis of the database mining. As it stands, this paper simply does not have a convincing case to be made.

R: We highly appreciate the above suggestions and comments. Through a thorough consideration, we realized that the abruptly increasing amounts of alkali basalts should be associated with the change in mantle temperature, rather than the back-arc extension. The continuous subduction recycled the increasing volume of oceanic crust into the mantle and enhanced the mantle cooling. Please see the revised manuscript. Thank you.

Reviewer #2 (Remarks to the Author):

Liu and coauthors present a statistical analysis of a large dataset of whole-rock geochemistry of mafic samples spanning some 3.5 Gyr.

The authors propose two specific endmember modes of "hot" (perhaps episodic) vs "cold" (modern) subduction, and propose a transition from the former to the latter has occurred over the past 3 Gyr. The proposed evidence for this shift is an increase in the abundance of alkali basalts (identified on the basis of high HFSE and P contents) over the past 3 Gyr.

In particular, the authors propose that alkali basalts act as a tracer of continental rifting driven specifically by slab rollback. They further claim that hot subduction cannot sustain slab rollback, and that this in turn explains the "scarcity of continental alkali basalts" in the geologic past.

While a transition from "hot" to "cold" subduction over the past 3 Gyr is broadly plausible (perhaps even tautologically inevitable) given secular mantle cooling, there is at least one, rather glaring, problem with the specific narrative presented here: of the classic continental rifts active on Earth today (East African, Rio Grande, and Baikal), not a single one is driven by, or even significantly associated with, slab rollback.

While the authors highlight several other localities where alkali basaltic magmatism may occur in a back-arc setting, the line from slab rollback to alkali basaltic magmatism is strained and nonunique; any type of low-degree mantle decompression melting will suffice to produce alkali basalts. Further, while we may expect some alkali basaltic magmatism in back arcs, back-arc magmatism typically displays a mixing between decompression and flux melting signatures, thus complicating the high HFSE enrichment expected for pure decompression-

derived alkali basalts by potentially mixing in an arc HFSE depletion signature. The use of high HFSE as a proxy for slab rollback is thus rather bizarre.

Concerningly, the criteria used as characteristics of continental alkali basalts (high HFSE and P contents, along with low silica) are more specifically signatures of low-degree mantle decompression melting, regardless of the tectonic setting. Consequently, we ought to observe an increase in alkali basaltic magmatism as a direct consequence of secular mantle cooling, independent of tectonic style.

R: We highly appreciate the above suggestions and comments. We realized that the abrupt increasing magnitude of alkali basalts should be associated with the change in mantle temperature, rather than the back-arc extension. So, we have made a major revision to the manuscript. The sharp increase of Diff (HFSE) at ca. 2.1 Ga reflect a rapidly increasing amounts of alkali basaltic rocks (from 5% to 15%), which is likely to be caused by an enhanced mantle cooling event rather than a gradual cooling process (also see Condie et al. 2016. *Geoscience Frontiers*). We thought out a better interpretation for rise of alkali basalts at 2.1 Ga. The continuous subduction recycled increasing volume of cold oceanic crusts into the mantle and enhanced the mantle cooling. By contrast, the episodic subduction in the stagnant-lid dominated tectonic regime before ca. 2.1 Ga could only decrease the mantle temperature at a low rate, because the episodic subduction often underwent temporary cessation or slow-down due to the frequent break-off of the subducting slab. Therefore, a transition from episodic to continuous subduction would dramatically accelerate the mantle cooling, and therefore resulted in lower degree of mantle melting and increasing amounts of alkali basalts. The proposal of an enhanced mantle cooling and style change of plate subduction around 2.1 Ga is also consistent with many previous studies (please see Discussion section in the manuscript).

A more direct interpretation of the dataset presented here would be that it confirms our expectation that mantle potential temperature and melting extent

have declined over the past 3 Gyr – which, though noteworthy, may not yield the sort of novelty or impact factor the authors are apparently hoping for.

R: We agree with the reviewer that mantle potential temperature and melting extent have declined over the past 3 Gyr.

However, the sharp increase of Diff (HFSE) at ca. 2.1 Ga reflect a rapidly increasing amounts of alkali basaltic rocks, which is likely to be caused by an enhanced mantle cooling event rather than a gradual cooling (Condie et al. 2016. *Geoscience Frontiers*). We suggest that the enhanced mantle cooling at ca. 2.1 Ga is attributed to the initiation of continuous plate subduction, because the continuous subduction could recycle increasing volume of oceanic crusts into the mantle. In contrast, an episodic subduction style before 2.1 Ga might yield frequent slab break-off during the subduction process due to the weak strength of oceanic plate in a hotter Earth. The slab break-off would lead to a temporal cessation or slowdown of the oceanic plate subduction, as revealed by numerical modelling (O'Neil et al., 2007. *EPSL*; van Hunen et al., 2008. *Lithos*; Moyen and van Hunen, 2012. *Geology*). That means the episodic subduction could not decrease the mantle temperature efficiently. Therefore, a transition from episodic to continuous subduction would dramatically accelerate the cooling rate.

We also found many geological evidences indicating the enhanced cooling of the mantle and the tectonic change at ca. 2.1 Ga. (1) According to the thermal evolution study results (Korenage, 2013. *Annual Review of Earth and Planetary Sciences*), the mantle temperature before 2.1 Ga had been >175 °C higher than its present temperature, exceeding the threshold of stabilizing a continuous subduction (Sizova et al. 2010. *Lithos*; Brown, 2014. *Geoscience Frontiers*). (2) The drop of mantle temperature at ca. 2.1 Ga (or 2.2-2.1 Ga) has also been observed by some other researches. Condie et al. (2000. *American Journal of Science*; 2016, *Geoscience Frontiers*) observed a MgO drop in komatiites and a decrease of magma generation temperatures (T_g) of basalts derived from depleted mantle during 2.5-2.0 Ga and interpreted them as an enhanced cooling of the mantle by subduction. (3) We calculated the proportion of rocks formed by high degree of mantle melting

(high MgO, low Rittmann Index) through time and also noticed a rapid decrease of degree of mantle melting at ca. 2.1 Ga (Supplementary Figure 5). (4) Spencer et al. (2018. *Nature Geoscience*) linked the magmatic flare-up subsequent to the magmatic lull (2.4-2.2 Ga) with the dramatic growth of continental crust and the transition from supercraton to supercontinent cycle. The supercontinent cycle is driven by continuous plate convergence that eventually lead to continental collisions and supercontinent assembly. The onset of continuous subduction exactly coincided with the start of a global-scale ca. 2.1-1.8 Ga collisional orogeny (Zhao et al., *Earth-Science Review*), followed by the formation of Nuna supercontinent. (5) The earliest ophiolites formed during 2.1-1.8 Ga. Although there are no such ophiolite records during the whole Mesoproterozoic, Cawood and Hawkesworth (2014. *Geology*) have suggested that the general paucity of passive margins through the Mesoproterozoic and the absence of any evidence for orogenesis argue for no significant breakup and reassembly of these continental fragments in the transition from Nuna to Rodinia, which can also explain the lack of ophiolites during the Mesoproterozoic. Considering the above geological evidences, we think that the rapid rise of alkali basaltic rocks at ca. 2.1 Ga reflected an enhanced cooling of the mantle, as a consequence of the initiation of continuous plate subduction.

Reviewer #3 (Remarks to the Author):

Please see my detailed review, attached as a pdf.

Review of Liu et al.: **“Hot-cold plate subduction transition marked by the rise of alkali magmatism ~2.1 billion years ago”**

I really enjoyed reading this paper, which presents statistical evidence in support of the widespread development of continental alkali basalts from 2.1–1.8 Ga. The authors provide convincing evidence that these types of rocks form in backarcs on the modern Earth, and link their widespread development in the Paleoproterozoic to continuous (or at least long-lived) subduction rollback that was likely made possible due to secular cooling of the mantle. The authors conclude that ~2.1 Ga marks the commencement of ‘modern-style cold subduction’.

The paper is generally well written and presents a smart but simple way of thinking about the data. It deals with exactly the sorts of things that interest me, and which are clearly of broad interest to the wider community. How, why and when plate tectonics became the dominant geodynamic ‘mode’ is a hot topic. I think this paper provides some strong support of a major change in the Paleoproterozoic, but I am not convinced that it is the change to ‘modern-style cold subduction’, at least not as far as I (and I would content many others) understand the term.

R: We highly appreciate the suggestion by the reviewer. We agree that the modern-style cold subduction is more suitable for the plate tectonics in the Neoproterozoic and Phanerozoic, which is characterized by the widespread low dT/dP metamorphic rocks. We prefer the term “continuous subduction”, as suggested by the reviewer below.

The main problem with the paper is a complete disregard of anything other than

'hot subduction' as the main tectonic mode on Earth between 2.1 and 3.5 Ga. Others have made arguments for a dominance of stagnant lid tectonics for much or perhaps all of the Archaean that gets not a single mention (see, for example, Cawood et al 2018 for some of the evidence: Cawood, P.A., Hawkesworth, C.J., Pisarevsky, S.A., Dhuime, B., Capitanio, F.A. and Nebel, O., 2018. Geological archive of the onset of plate tectonics. *Philosophical Transactions of the Royal Society A: Mathematical, Physical and Engineering Sciences*, 376).

R: Agreed. We only focused on the change of plate subduction style but forget to discuss it with the tectonic regime. The episodic subduction should occur in the stagnant-lid dominated vertical tectonics. We have revised the manuscript accordingly. Thanks.

As noted recently by Brown and Johnson (Secular change in metamorphism and the onset of global plate tectonics. *American Mineralogist*, 103, 181–196, 2018.), which uses a much larger metamorphic database than Brown (2006), and might warrant a mention, the 'modern-style' of subduction, representing cold collision and deep subduction of the slab prior to breakoff, is associated with the formation of low dT/dP rocks, most conspicuously blueschists. While there are rare examples of such rocks as old as ~2.1 Ga, as the authors note, they do not appear widely until the Neoproterozoic (~0.75 Ga). This is starkly illustrated below, using an updated database comprising 564 data points from a paper (Brown, M., & Johnson, T. (2019). Metamorphism and the evolution of subduction on Earth. *American Mineralogist*, in review) that I will try to remember to attach.

Of course, the authors might argue that this is a preservation issue, and that the much older (Paleoproterozoic) data are the only remaining examples of low T/P rocks that were much more widespread. However, it seems more likely to me that they record localised examples of cold subduction, not (global) plate tectonics. In my view, arguing for the onset of global ‘modern’ subduction would require much more support than here. In any case this needs much more discussion (see also below).

R: We agree that the term “modern-style subduction” should be more suitable for the subduction in the Phanerozoic and Neoproterozoic. We like to use the term “continuous subduction” for the subduction style during the late Paleoproterozoic and the whole Mesoproterozoic. Actually, the word “continuous” is more appropriate to interpret the findings from Fig. 4. The mantle temperature before 2.1 Ga was too hot to produce alkali basaltic rocks. The sharp increase of Diff (HFSE) at ca. 2.1 Ga reflect a rapidly increasing amounts of alkali basaltic rocks (from 5% to 15%), as a result of the decreasing degree of mantle melting. This is most likely to be caused by an enhanced mantle cooling event rather than a gradual cooling process (also see Condie et al. 2016. *Geoscience Frontiers*). We suggest that the enhanced mantle cooling at ca. 2.1 Ga is attributed to the initiation of continuous plate subduction, because the continuous subduction could recycle increasing volume of oceanic crusts into the mantle. In contrast, an episodic subduction style before 2.1 Ga might yield

frequent slab break-off during the subduction process due to the weak strength of oceanic plate in a hotter Earth. The slab break-off would lead to a temporal cessation or slowdown of the oceanic plate subduction, as revealed by numerical modelling (van Hunen et al., 2008. *Lithos*; Moyen and van Hunen, 2012. *Geology*). That means the episodic subduction could not decrease the mantle temperature efficiently. Therefore, a transition from episodic to continuous subduction would dramatically accelerate the cooling rate.

What the new metamorphic data do show is the emergence of 'paired metamorphism' sometime in the late Archean to Paleoproterozoic. The Neoproterozoic metamorphic data of Brown & Johnson (2018, and in review) can be described by a unimodal Gaussian distribution. Sometime in the Paleoproterozoic, the data exhibit a distinctly (and statistically significant) bimodal character that becomes increasingly bimodal prior to the widespread appearance of blueschist that records the onset of (what we would argue is) 'modern' plate tectonics--see density plot below. This is something else we have currently in review (Holder et al.).

R: We agree that the onset of plate tectonics can be determined by the emergence of paired metamorphism. We are happy to do some more discussions with the reviewer on this topic. Thanks for providing the above two figures, which include the most recent discoveries of low dT/dP metamorphic rocks at 2.1-1.8 Ga and presented a very creative data processing method. An interesting

finding from the above contour figure is that we noticed a transition from unimodal Gaussian distribution in the late Archean to bimodal distribution in the Paleoproterozoic, like the reviewer pointed out. The reviewer also drew two arrows to emphasize it. Although widespread records of paired metamorphism started to appear in the Neoproterozoic (Brown and Johnson, 2018), the high dT/dP and intermediate dT/dP metamorphic groups become more divergent in the Paleoproterozoic, around ca. 2.2 Ga and stabilize in the Proterozoic (also see figure below). Paired metamorphic belts are diagnostic of plate subduction with the low T/P (blueschist-bearing) belt lying next to the trench and the high T/P (greenschist) belt near the arc (Stern, 2008. *Geology*). Can we speculate that the divergence of high dT/dP and intermediate dT/dP metamorphism (bimodal distribution) since ca. 2.2 Ga may indicate the start of continuous subduction, while the relatively less divergent high dT/dP and intermediate dT/dP metamorphism may correspond to the episodic subduction? Brown and Johnson (2018. *American Mineralogist*) also noticed a rise in thermal gradients of high dT/dP metamorphism around 2.3-2.1 Ga and interpreted it as the reconfiguration from supercratons to the first supercontinent (see figure below). The supercontinent cycle should be driven by continuous plate convergence that finally lead to continental collisions and supercontinent assembly. Of course, more investigations on those metamorphic rocks are required to support this speculation.

To summarize, I think the authors are identifying the onset of continuous/long-lived subduction and rollback at 2.1 Ga (i.e., the onset of plate tectonics, a globally-linked network of narrow plate boundaries) from an earlier regime dominated by stagnant lid tectonics and/or shallow/failed/hot subduction. I do not think ~2.1 Ga it is the establishment of the modern regime, which is characterized by low T/P metamorphism and the widespread formation of blueschists. Again, the authors are under no obligation to agree, but they do need to discuss these possibilities (that are already published).

R: Thanks very much for the suggestions. We also believe that the continuous subduction is more appropriate to describe the subduction style after ca. 2.1 Ga. We agree that the modern regime should be characterized by low T/P metamorphism and the widespread formation of blueschists.

All in all, I think this is a strong paper with some significant findings. However, I would personally rethink the implications of the data presented here within the context of the existing metamorphic and other data. The paper could be expanded by a quarter to a third to discuss several aspects in better details. I believe Nature Comms permits longer (>4 pages) format papers, but the appropriate link was broken when I tried it:

http://www.nature.com/ncomms/about_journal.html

R: We are very happy that the reviewer is interested in our study. We hope our findings can be helpful to the reviewer in his future works. We believe that the increase of alkali basalts is reconciled to the bimodal distribution of T/P of metamorphic rocks in the Neoproterozoic, although we have not mentioned it in the manuscript as that figure has not been published.

We have expended the paper length. Thank you.

Some specific comments keyed by line number follow:

Lines 57–60: Yes, but what is the evidence that these examples record global rather than local episodes?

R: We agree that they are local episodes. We have modified the statement in the manuscript.

73–74: This sentence is a bit clumsy and could be redrafted.

R: Thank you. This sentence and several clumsy sentences have been removed or redrafted.

125–128: I disagree that most previous studies agree what is stated. To reiterate, there is no absolutely no mention of ‘stagnant lid’ (thick plateaux/‘vertical’ tectonic/sagduction/etc.) models that many believe are perfectly plausible. This is a major shortcoming of the paper.

R: Agreed. Originally, we only keep an eye on the style change of subduction. We forgot to discuss the episodic subduction in its stagnant-lid tectonic regime. This problem has been fixed in the whole manuscript.

131: I don’t understand the meaning of ‘differentiation trends’. Unless I’m mistaken, the compositional signatures you are describing are a function of the

source not of magmatic fractionation. Is 'secular trends' or 'compositional trends' better?

R: Agreed. We have changed the terms into compositional trends. Thank you so much for this suggestion.

142–144: Or they reflect a change from a pre-plate tectonic regime to a plate tectonic regime. No problem if you disagree, but the evidence needs to be discussed.

R: Actually, we agree that this is a change from a stagnant-lid dominated vertical tectonics to plate tectonics. However, there are a lot of different opinions on the onset of plate tectonics. Somebody may also consider the episodic subduction as a component of plate tectonics. Besides, there is still no definite criteria for the termination of stagnant-lid tectonics. Maybe it is a progressive emergence of plate tectonics rather than a sharp boundary between the stagnant-lid tectonics and plate tectonics. In the late Archean and the early Paleoproterozoic, plate tectonics might have proceeded in episodic subduction but the stagnant-lid mantle convection might also exist. As such, we just use the term “continuous plate subduction” to describe the subduction style after 2.1 Ga. Again, thank you very much for this suggestion.

159–152: Definitely correct. Might this warrant a reference or two?

R: References have been cited at appropriate places in this whole section. Thanks for this suggestion.

185–186: There is a clear change here, but there are others (notably the big peak at ~1.2 Ga and the low at 0.6 Ga). I assume these are interpreted in terms of the supercontinent cycle, but some more details would be good...

R: Through checking the raw data, we found that most of those variations are caused by samples from diabase/dolerite dyke, of which the incompatible

element concentrations are strongly affected by crustal contamination (see Methods). Therefore, we removed all the diabase/dolerite dyke samples from the dataset and re-calculate the Diff (HFSE). Now, there are no large fluctuations of Diff (HFSE) between 1.8 and 0.1 Ga (Figure 4a). The abnormally high Diff (Nb) and Diff (P) for the latest 0.1 Ga are caused by sampling bias, which have been explained in the manuscript (Results section) and Supplementary Figure 3. It seems there is a low-value range of Diff (HFSE) between 0.9 and 0.5 Ga, but those values are still above zero and the Diff (Nb), Diff (Ti) and Diff (P) do not vary together. Thus, we think it is not worthy of discussion.

201–203: ...as the variations do not seem to have the same episodicity as the metamorphic or detrital (and other) zircon record.

R: Similar to the above explanation, the fluctuation disappeared after removing dolerite/diabase dyke samples.

277–278: To reiterate, I have big problems with the assumption that ‘hot subduction’ was the main or only game in town from 3.5 to 2.1 Ga. Some regard needs to be paid to alternative models.

R: Agreed. We have changed all the expression “hot subduction” into “episodic subduction” and also mention it together with its stagnant-lid tectonic regime.

Fig. 4. I may have missed it, but some details as to how the curves in (a) were calculated would be helpful. I assume they are a moving mean, but for what window size and what step?

R: Agreed. The moving average smoothing has been introduced in the Method section. We also added some details in the figure caption. The window size is 0.5 Ga, while the step width is 0.1 Ga. The trends are comparable to the 0.2-Ga binned secular variations of Diff (HFSE) (Supplementary Figure 6).

A couple of refs for 'sagduction' models:

Bédard, J.H., 2006. A catalytic delamination-driven model for coupled genesis of Archaean crust and sub-continental lithospheric mantle. *Geochimica et Cosmochimica Acta*, 70(5), pp.1188-1214.

Johnson, T.E., Brown, M., Kaus, B.J. and VanTongeren, J.A., 2014. Delamination and recycling of Archaean crust caused by gravitational instabilities. *Nature Geoscience*, 7(1), p.47.

R: Thanks a lot. We have cited the above references when we describe the stagnant-lid tectonic regime in the text.

REVIEWERS' COMMENTS:

Reviewer #1 (Remarks to the Author):

The authors have thoroughly revised the manuscript and now adopt an interpretation that reflects many of the comments in my earlier review. This paper now presents a convincing and interesting observation and ties it in well with other approaches and ideas concerning the onset of plate tectonics and deep subduction and its consequences for an evolving Earth. While much of the discussion of this period of Earth history is inevitably speculative, this paper makes a useful contribution showing a rather abrupt change in the compositional characteristics of igneous rocks through time. It is a useful contribution to the literature and should engender broad interest.

Charles Langmuir

Reviewer #2 (Remarks to the Author):

In this revision, the authors have made a good-faith attempt to respond to the concerns of three reviewers including myself. Unfortunately from my perspective, this has included incorporating a number of opinions from reviewer 3 that I find misguided, ill-supported, and reliant on a one-sided reading of the literature. Nonetheless, they have largely removed the major flaw in the original manuscript (evidently identified by both reviewer 1 and myself) regarding the interpretation of alkali basalts as a diagnostic signature of back-arcs and slab rollback.

Reviewer #3 (Remarks to the Author):

In my opinion, the authors have done a good and careful job in addressing my comments, and those of the other reviewers. They still argue that their data infer an 'abrupt' change in mantle potential temperature at c.2.1 Ga. The metamorphic data would argue more for a gradual decline T_p (IN CONFIDENCE see the attached Holder et al. paper, which has now been accepted for publication in Nature pending approval of all the signed copyright forms, etc.). A comment on this might be

welcome if the author see fit. However, presenting data in support of plausible alternative models will hopefully stimulate further research into the early Earth, and is very welcome.

As one small suggestion, for clarity I would change the sentence in the abstract from:

"While plate subduction proceeds continuously today, episodic subduction predominantly occurred in the early stagnant-lid tectonic stage.episodic subduction predominantly occurred in the early stagnant-lid tectonic stage."

to something like

"While continuous subduction and an interconnected network of narrow plate boundaries are features of modern plate tectonics, a stagnant-lid tectonic regime with localised episodes of subduction likely characterized the early Earth."

Tim Johnson

A point-by-point response

Here is our point-by-point response to the reviewers' comments.

Reviewer #1 (Remarks to the Author):

The authors have thoroughly revised the manuscript and now adopt an interpretation that reflects many of the comments in my earlier review. This paper now presents a convincing and interesting observation and ties it in well with other approaches and ideas concerning the onset of plate tectonics and deep subduction and its consequences for an evolving Earth. While much of the discussion of this period of Earth history is inevitably speculative, this paper makes a useful contribution showing a rather abrupt change in the compositional characteristics of igneous rocks through time. It is a useful contribution to the literature and should engender broad interest.

Charles Langmuir

Reply: Thanks for your previous comments and this commendation.

Reviewer #2 (Remarks to the Author):

In this revision, the authors have made a good-faith attempt to respond to the concerns of three reviewers including myself. Unfortunately from my perspective, this has included incorporating a number of opinions from reviewer 3 that I find misguided, ill-supported, and reliant on a one-sided reading of the literature. Nonetheless, they have largely removed the major flaw in the original manuscript (evidently identified by both reviewer 1 and myself) regarding the interpretation of alkali basalts as a diagnostic signature of back-arcs and slab rollback.

Reply: Thanks for the previous comments. While many literatures on the interpretations of tectonic change during the Paleoproterozoic are speculative, it requires more efforts in the further works. Our study provides an important evidence to the change of styles of plate subduction and will engender broad interest.

Reviewer #3 (Remarks to the Author):

In my opinion, the authors have done a good and careful job in addressing my comments, and those of the other reviewers. They still argue that their data infer an 'abrupt' change in mantle potential temperature at c.2.1 Ga. The metamorphic data would argue more for a gradual decline T_p (IN CONFIDENCE see the attached Holder et al. paper, which has now been accepted for publication in Nature pending approval of all the signed copyright forms, etc.). A comment on this might be welcome if the author see fit. However, presenting data in support of plausible alternative models will hopefully stimulate further research into the early Earth, and is very welcome.

Reply: Thanks for this information on the gradual decline of T_p . I have a strong feeling that the tectonic change in the Paleoproterozoic will be a hot topic in the coming years. The paper by Holder et al. has not been published, but another recent paper by Sobolev and Brown (2019) in Nature presented very interesting observations on the "lubrication events". The first subduction lubrication event after the 2.45-2.2 Ga Huronian global glaciations stabilized the plate subduction and triggered the assembly of the Columbia Supercontinent (Sobolev and Brown, 2019), which is generally coincident with the onset of continuous subduction identified by us. Thus, it is believed that the cooling pace of the mantle was, more or less, accelerated after 2.2-2.1 Ga. I think whether this change is abrupt or gradual may depend on the methods adopted by the researchers.

As one small suggestion, for clarity I would change the sentence in the abstract from:

"While plate subduction proceeds continuously today, episodic subduction predominantly occurred in the early stagnant-lid tectonic stage.episodic subduction predominantly occurred in the early stagnant-lid tectonic stage."

to something like

"While continuous subduction and an interconnected network of narrow plate boundaries are features of modern plate tectonics, a stagnant-lid tectonic regime with localised episodes of subduction likely characterized the early Earth."

Tim Johnson

Reply: Thanks very much for this suggestion. We have modified this sentence. Since there is a word limit of the abstract in accordance with the journal's instruction, we have deleted some words. This sentence is now modified into "While continuous subduction is a typical feature of modern plate tectonics, a stagnant-lid tectonic regime with localized episodic subduction likely characterized the early Earth."